# Predicting Determinants of Lifelong Learning Intention Using Gradient Boosting Machine (GBM) with Grid Search

**Chayoung Kim** [1] and **Taejung Park** [2,*]

1 College of Liberal Arts and Interdisciplinary Studies, Kyonggi University, 154-42, Gwanggyosan-ro, Yeongtong-gu, Suwon-si 16227, Gyeonggi-do, Korea; kimcha0@kgu.ac.kr
2 Department of Life-Long Education & Counseling, College of Future Convergence, Pukyong National University, 45, Yongso-ro, Nam-gu, Busan 48513, Korea
* Correspondence: edutechpark@gmail.com; Tel.: +82-51-629-5907

**Abstract:** The purpose of this study is to explore the factors that have the most decisive influence on actual learning intention that leads to participation in adult education. For developing the predictive model, we used tree-based machine learning, with the longitudinal big data (2017~2020) of Korean adults. Based on the gradient boosting machine (GBM) results, among the eleven variables used, the most influential variables in predicting the possibility of lifelong education participation were self-pay education expenses and then highest level of education completed. After the grid search, not only the importance of the two variables but also the overall figures including the false positive rate improved. In future studies, it will be possible to improve the performance of the machine learning model by adjusting the hyper-parameters that can be directly set by less computational methods.

**Keywords:** lifelong learning intention; machine learning; gradient boosting machine (GBM); grid search

## 1. Introduction

For a long time, large number of studies on factors influencing adult lifelong learning intention and participation have been conducted [1]. Educational background, competency, gender, occupational status, and occupational characteristics have been variously explored as determinants of lifelong learning participation [2–4] but have not shown consistent results. The reason may be the small number of target groups analyzed in each study, different characteristics of the target groups, or differences in the time or period of the study. In addition, it may be due to the fact that the variables included in the research model are different, or it may be due to the fact that the type of lifelong learning that is the dependent variable is different [5]. Therefore, it is necessary to re-explore the factors that have a decisive influence on lifelong learning participation in reality based on the massive data systematically collected in the form of a longitudinal study.

Recently, machine learning algorithms have been proposed for prediction in various fields, and have achieved remarkable results. Due to the recent rapid development of computer specifications and the development of machine learning libraries, research on predictive model development using machine learning techniques is being actively conducted in various fields including education [6]. Therefore, this study intends to propose a prediction method based on classification learning, a machine learning technique, to predict participation in lifelong learning. A prediction technique using the Gradient Boosting Machine Learning Algorithm was proposed to predict lifelong learning participation by using the 2017–2020 longitudinal study data of the Seoul Lifelong Learning Survey, Republic of Korea. The results of this study are expected to deepen the existing understanding of lifelong learning since it is possible to accurately predict participation in lifelong learning and can be used as important basic information to promote participation in lifelong learning in the future.

The intention to engage in a particular behavior is a strong determinant for actual behavior [7]. For instance, intention to engage in learning activities was a robust predictor of actual participation [8]. In other words, being involved in learning activities is a first meaningful step towards actually engaging. Rainbird [9] emphasizes the importance of both individual characteristics, and factors that are external to the individual in relation to the individual's characteristics and structural position within the organization. Baert, De Rick, and Van Valckenborgh [10] also acknowledge that factors influencing this decision-making process in the theoretical model are related not only to the characteristics of the learner, but also to the characteristics of learning activities and social contexts. Baert and other colleagues [10] put the development of learning intentions at the center of the decision-making process of potential learners. This decision-making process is regarded as the improving clarification of educational needs [11].

The process begins with a need, a recognition that something is lacking or that a contradiction exists between someone's current situation and the desired situation. This requirement can include many aspects such as money, time, resources, materials, health status, etc. When the educational aspect is recognized, the educational need becomes more or less clear. For example, the birth of a third child leads to the perception that someone's current home becomes too small as the child grows. This can lead to the fact that this person requires a promotion and wants to earn more money to buy or rent a larger house. While realizing this, in order to earn that promotion, he may need to expand his knowledge and skills [4].

According to the theory of rational behavior by Ajzen and Fishbein [12], learners who articulate educational needs should develop intentions to engage in learning. The development of this learning intention can arise from the expression of a specific learning intention conceptualized by Confessore and Park [13] as a basic element of autonomy. It assumes that the individual has an intrinsic or extrinsic motivation to learn. Ajzen and Fishbein [12]'s sociopsychological model emphasizes the intentional nature of decision makers [14], and this intention leads to participatory behavior of people, leading to the formation of educational demands [10]. For example, learners search for educational programs and self-learning activities that help meet their needs. If he finds one, this does not automatically mean that the found solution is also the best or most suitable solution. The purpose of the program may be tailored to his interests, but the duration, lessons, location or price of the program present obstacles to effective participation. After evaluating all relevant functions, the final step in the decision-making process is to engage in actual training [4].

Lifelong learning is a continuous support process that motivates and empowers individuals to learn and develops human potential [15] and is used in a way that is compatible with lifelong education, social education, adult education. A learner's decision to participate in educational activities is not a single action but a result of a series of actions or chain reactions [16]. The main factors influencing the decision to participate in education are gender, life cycle, educational background, income level, and marital status which are characteristics of personal background. Darkenwald and Merriam [17] emphasized that the main factor influencing adult learners' participation decisions in lifelong learning is the social environment to which an individual belongs, especially socioeconomic status. According to the results of the OECD [18] survey, time management factors such as lack of time (36.1%), too much work (17.4%), and having to take care of the family (15.4%) were the most common reasons for not participating in job-related lifelong learning. The next most common factors corresponding to the characteristics of training were cost (23.3%), lack of a favorite course (12.8%), and lack of time frame (12.2%) and the most common reason for personal circumstances was lack of support from employers (10.2%). Looking at the status of Korean adult lifelong learning, it is said that the factors determining whether to participate in lifelong learning tend to increase with higher income, higher education, and younger age [19].

Unlike previous studies that focused only on the individual characteristics of learners, studies such as Darkenwald [20], Ivy [21], and Parasuraman [22] have shown that that the factors participating in lifelong learning are influenced not only by the psychological and environmental characteristics of individuals but also by the influence of the participating lifelong education institutions. Based on research by Emmalou Van Tilberg [23], the participation of farmers in education includes economic and physical factors such as the cost of participating in the training, the distance to the place of participation and the time to reach the place, institutional factors related to reliability of educational institutions, reliability of instructors, educational content, training delivery method, appropriateness of training time, training program design and operation, and other factors include training participants' education level, age, gender, and income. Ivy [21] presented educational institution facilities, accessibility, support environments, and staff services as factors that influence adult learners' participation in education. A recent study of Thongmak [24] on the effects of individual, institutional, and pedagogical factors on lifelong learners' intervention and motivation and intention recommend institutions and companies to expose learners to media regarding to the critical core skills to improve their learning intention.

Taken together, these findings reveal the importance of both personal, institutional, and socio-contextual influences on the lifelong learning intention and participation of individuals. Baert and other colleagues [10] classified the variables affecting participation in lifelong learning into three categories such as micro-level, meso-level, and macro-level based on the original Bronfenbrenner [25,26]'s ecological system theory of the micro-system, meso-system, and macro-system contexts of human development. According to the Bronfenbrenner's revised theory proposes that the person's development takes place through the process of progressively more complex reciprocal interactions between an active biopsychological human organism and the persons, objects, and symbols in its immediate external environment [27]. Park and Cha [28]'s support vector machines with recursive feature elimination (SVM-RFE) analysis results, Korean high-school graduates on the decisions of choosing and newly entering universities were mostly affected by the mesosystems of interactions with parents, while re-enrollers were affected by the macrosystems of social awareness as well as individual estimates of talent and aptitude of microsystems. To understand the factors influencing lifelong learning in the context of the lifelong learners' interactions with their environments, this study also divided the environment surrounding learners' lifelong learning intentions into three ecological levels. The three levels are as shown in Table 1: the perception about the characteristics of the learner (micro-level), the characteristics of the institutional programs and learning activities (meso-level), and the broader social context and its actors (macro-level). These factors determine the attitude of the individual, and consequently influence the development of the learning intention of that individual.

**Table 1.** Factors influencing the lifelong learning intention.

| Level | Factors |
|---|---|
| Micro-level | Characteristics of the Learner (Individual Characteristics)<br>- Socio-Demographic Characteristics<br>- Psychological Characteristics<br>- Characteristics Regarding the Living Situation<br>- Characteristics Regarding Learning, Education and Training |
| Meso-level | Characteristics of the Institutional Programs and Learning Activities |
| Macro-level | Characteristics of the Social Context and its Actors |

## 2. Methods

### 2.1. Boosting-Based Machine Learning

The biggest problem that appears when applying machine learning techniques is overfitting. The boosting technique trains the datasets on a tree basis, but unlike the

bagging technique where each tree is independent of each other, the technique is vulnerable to overfitting by learning the tree by weighting the parts with large sequential errors. The overfitting problem can be prevented by optimizing various hyperparameters of the boosting models. A representative machine learning technique based on such boosting is gradient boosted regression trees (GBRTs), which is the latest gradient boosting ensemble technique. However, GBRTs take a considerable amount of calculation time to train the model, and several algorithms have been developed to supplement this. Among them, the light gradient boosting machine (LightGBM) algorithm is typically a model in which the training time is considerably shortened [29].

The boosting method during classification learning is one of the techniques for generating multiple classifiers by manipulating initial sample data similar to bagging, but the biggest difference is that it is a sequential method. The boosting technique is a method of adjusting the sample weight of the training data of the next classifier based on the training result of the previous classifier to proceed with the training [30].

As shown in Figure 1 in the boosting method, training data and test data are randomly extracted at an appropriate ratio and divided. Then, samples are extracted from the test set using the boost trap sampling technique and applied to a specific learning algorithm to create a classifier. Through the classification result of the classifier generated in this way, weights are given to the misclassified data and the unextracted (not used for learning) data to be used for the next learning. This series of processes is called a boosting round. In this way, a final classification model is created using the completed models through a total of "n" boosting rounds [31].

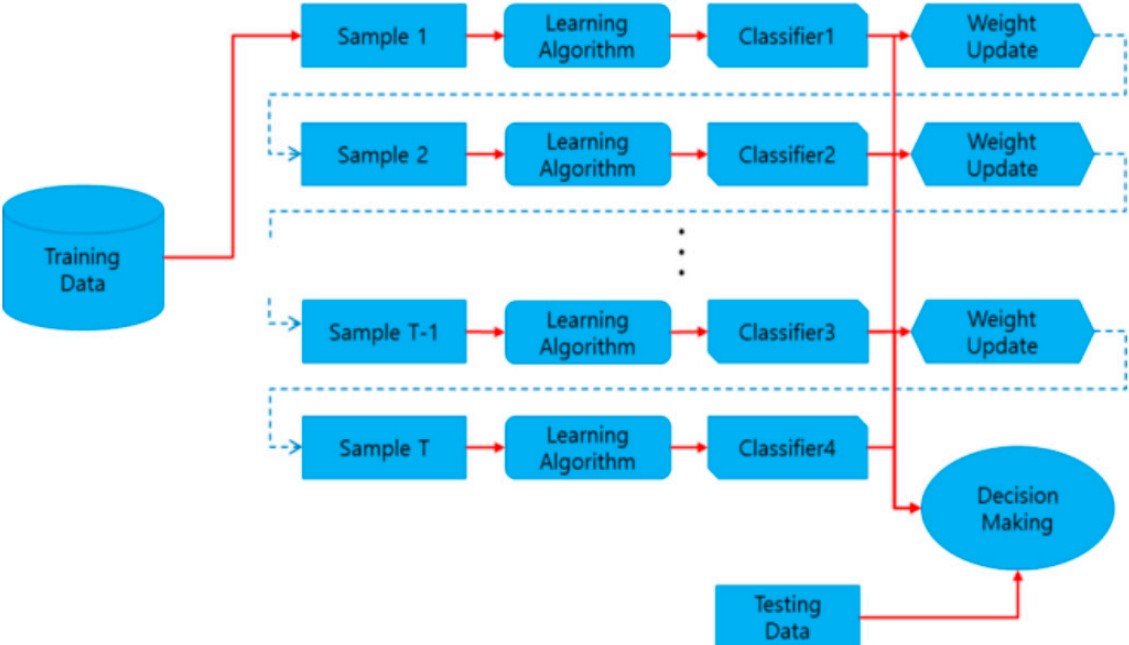

**Figure 1.** Boosting flow chart ([30,31]).

Boosting algorithm, one of the machine learning algorithms, plays an important role in handling bias-variance-tradeoff. Unlike the bagging algorithm, which controls only the high variance in the model, boosting is considered to be more effective since it controls both aspects (deflection and variance). Boosting is a sequential technique that works according to the principles of the ensemble and combines a weak set of learners to provide improved prediction accuracy. The model result at any moment t is weighted based on the result at the previous moment t − 1. Correctly predicted results are given lower weights, and misclassified results are given higher weights. This technique is available for both classification and regression.

The easiest way to understand GBM is to understand it as residual fitting. If we predict y through a very simple model A, then predict the remaining residuals again through model B, and predict y through model A + B, we can make a better model B than A. If we continue in this way, the residuals will continue to shrink, and we will be able to build a predictive model that describes the training set well. However, this method has the disadvantage that although bias can be significantly reduced, overfitting may occur. If you look at Figure 2 you can see that the residuals that are predicted through tree1 are predicted through tree 2, and the residuals are gradually reduced by repeating this. At this time, each model tree 1, 2, 3 are called weak learners, and a classifier combining them is also called a strong learner. A simple decision tree is often used as a weak classifier. This is also called a gradient boosting tree, and recently implemented representative libraries include LightGBM and XGboost. XGBoost is an algorithm that supplements the shortcomings of the existing GBM algorithm. GBM's gradient descent is a method to find the optimal parameter that minimizes the loss function. Gradient boosting boosts performance by focusing on other models when the gradient reveals the weakness of the model learned so far. However, gradient boosting is slow and has the problem of overfitting as mentioned above.

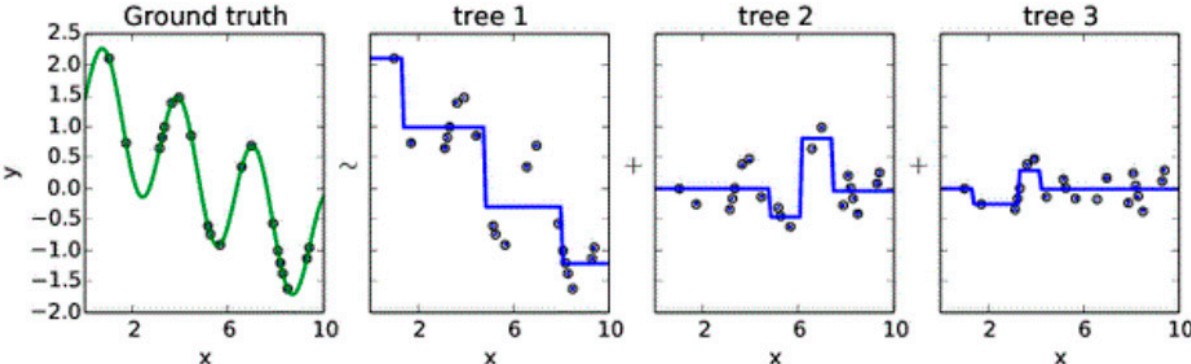

**Figure 2.** Gradient boosting example (Source: https://machinelearningmastery.com/gentle-introduction-gradient-boosting-algorithm-machine-learning/ (accessed on 28 May 2021)).

A common approach to parameter tuning is to tune two types of parameters which are (1) tree-based parameters and (2) boosting parameters. GBM is powerful enough not to overfit to tree growth, but for a given learning_rate a high number can lead to overfitting. Usually, a relatively high learning_rate is chosen. In general, the default 0.1 works well, but it may work well somewhere between 0.05 and 0.2. After that, n_estimators (optimal number of trees) are determined for the corresponding learning_rate. Usually around 40–70 range is suitable. After that, the parameters for each tree are adjusted for the determined learning_rate and the number of trees. After obtaining rational per-tree parameters, a more robust model is obtained by lowering the learning_rate and increasing the estimator proportionally [32].

XGBoost is a model born to complement this problem. XGBoost is able to solve real world scale problems using a minimal amount of resources [33]. XGBoost is a method introduced by Chen and Guestrin [33] for the purpose of solving overfitting problems in linear or tree-based models and improving the stability and training speed of large datasets. It is an abbreviation of eXtreme gradient boosting and is a boosting algorithm-based model, and is a flexible model that supports regression, classification, ranking, and user-defined objectives [30]. In the case of recent gradient boosting, it has excellent predictive performance, but it is difficult to tune the optimization model due to the disadvantage that it takes a long time to perform. However, as algorithms that shorten the execution time while improving the prediction performance of existing gradient boosting, such as XGBoost and LihgtGBM, continue to appear, it is the most useful algorithm in the classification of structured data and is receiving the most attention in boosting algorithms. It's not an overwhelming numerical difference, but it generally shows better predictive performance

than other machine learning methods in classification. XGBoost is based on the existing Gradient boosting since the weighting, which is a characteristic of ensemble boosting, is gradient descent, but it is faster than GBM and includes regulations such as early stopping to prevent overfitting. XGBoost has better performance than previous GBM, but still has the disadvantage of slow learning time. LightGBM came out to compensate for these shortcomings of XGBoost. LightGBM can handle large amounts of data, use less memory, and is fast, but it also has its drawbacks, which can lead to overfitting if too little data is used. Unlike the existing boosting models including XGBoost, lightGBM divides the tree around leaf nodes. Figure 3 shows the difference between the methods. Level-wise tree analysis has to be balanced, so the depth of the tree is reduced and operations are added. LightGBM continuously splits the leaf nodes in the direction that can reduce the loss regardless of the balance, so an asymmetric and deep tree is created, but loss can be reduced compared to level-wise when generating the same leaf. Therefore, LightGBM can be efficient if you have to deal with large datasets within a limited time. Recently, the problem of time tends to be solved with a large set of GPUs, so XGBoost is still widely used.

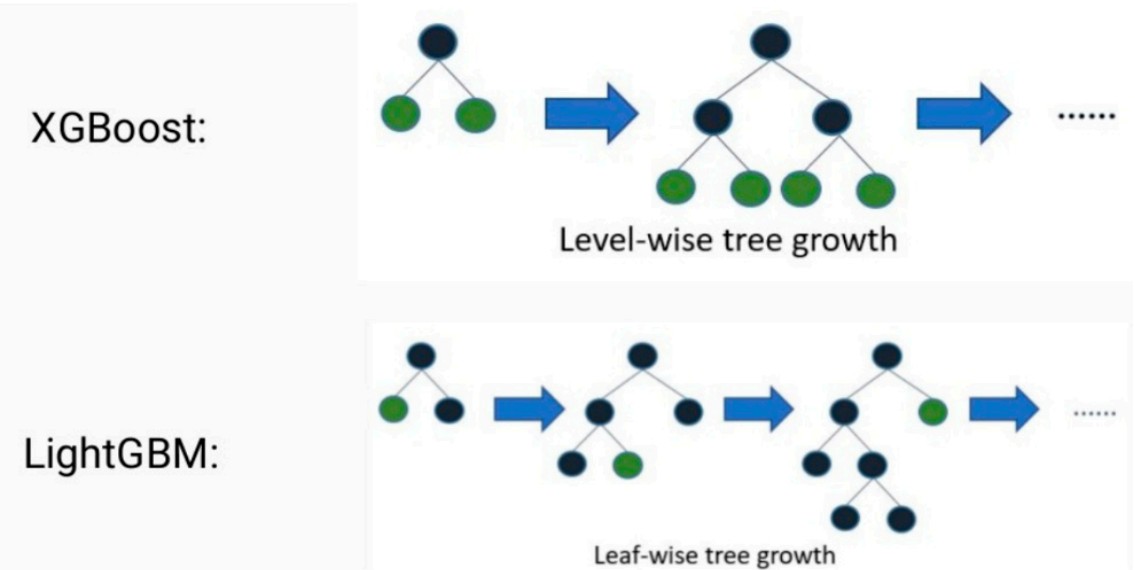

**Figure 3.** Difference between LightGBM and XGBoost (Source: https://rohitgr7.github.io/lightgbm-another-gradient-boosting (accessed on 13 July 2021)).

Yağcı [34] explored the hidden relationship of education data already known in education data mining and proposed a tool to predict students' academic achievement using a machine learning model. However, in Yağcı's study, nearest neighbors, support vector machines, logistic regression, Naïve Bayes, and k-nearest neighbor algorithms was utilized as a prediction method and GBM was used to classify high academic achievement. While Yağcı's [34] study used GBM as it is, we proposed a way to efficiently and effectively better hyperparameters of GBM.

### 2.2. Proposed Method

In this study, the GBM Python scikit-learn library, which is known as the most basic of boosting, was used, and Grid search was used to tune hyper-parameters. Therefore, after building a model with the most recent versions of GBM, GBRTs, LightGBM, and XGBoosting, it is expected that similar results will be obtained even if hyper-parameter tuning is performed using grid search or random search.

This study shows that even if the method provided by the Python library is used, much better results can be obtained by selecting a scientific method by the user to adjust the hyperparameters. Unlike the parameters set by the model or data, this study focused more on the hyper-parameters that the field user can set with insight. In other words, we

do not aim to improve the GBM method, but rather focuses on hyper-parameter setting based on the user's insight regardless of which method is used.

Data sets from 2017 to 2020 of the Seoul Lifelong Learning survey were used, and each feature correlation of big-data was confirmed at the data pre-processing stage. As shown in Table 2, 11 important characteristics could be selected. In Table 3, we have shown that one characteristic, employment type, was used as a target for the classification model. Figure 4 shows the feature ranking according to importance, which was obtained through simple regression analysis. According to this, academic background, self-pay tuition, and program type are the most important, and it was confirmed that there is no relation with variables such as gender or age that are traditionally considered to be related. The ranking of the importance is also changed by the gradient boosting and subsequent grid search. The ranking provided the basis for using the yes/no classification of self-paying learning expenses when performing the gradient boosting classification model.

**Table 2.** Features with high correlation.

| | | | |
|---|---|---|---|
| Micro-Level | B3D_10_IM | Self-Pay Learning Expenses—Vocational Competency Improvement Education (1) (after non-response substitution) | Learning expenses spent in the past year $\_\_\_\_\_\_\_\_\_\_\_\_\_\_\_\_ |
| | DQ1A | Highest level of education completed—school level (including non-response) | ① Uneducated, ② Elementary school, ③ Middle school, ④ High school, ⑤ University (2 or 3-year university), ⑥ University (4-year university), ⑦ Graduate school (Master), ⑧ Graduate school (Ph.D.) |
| | DQ7 | Main source of income | ① Earned by me, ② Interest and rental income, ③ Allowance money from family, relatives, and children, ④ Pension, ⑤ Subsidy, ⑥ Others |
| Meso-Level | B3D_3 | Program Type—Vocational Competency Improvement Education (1) | ① Lectures taught by instructors at a certain place, ② On-the-job training programs, ③ Remote/Cyber courses, ④ Professional seminars and workshops, ⑤ Study clubs, ⑥ Other lectures and private tutoring |
| | B3D_14 | Program Satisfaction—Vocational Competency Improvement Education (1) | ① Very dissatisfied, ② Dissatisfied, ③ Normal, ④ Satisfied, ⑤ Very satisfied |
| | B4_2 | Increasing psychological satisfaction and happiness | ① Not helpful at all, ② Not very helpful, ③ Medium, ④ Slightly helpful, ⑤ Very helpful, ⑥ Not applicable |
| | B4_3 | Self-development, such as cultivating culture and acquiring knowledge | ① Not helpful at all, ② Not very helpful, ③ Medium, ④ Slightly helpful, ⑤ Very helpful, ⑥ Not applicable |
| Macro-Level | B3D_11 | Whether external support for learning expenses is provided—Vocational Competency Improvement Education (1) | ① Yes, ② No |
| | B4_8 | Social Participation (Volunteer Service and Community/Social Activities) | ① Not helpful at all, ② Not very helpful, ③ Medium, ④ Slightly helpful, ⑤ Very helpful, ⑥ Not applicable |
| | J2_3 | Degree of improvement in quality of life by participation in lifelong learning—Satisfaction with social participation | ① Not helpful at all, ② Not very helpful, ③ Medium, ④ Slightly helpful, ⑤ Very helpful, |
| | J2_4 | Degree of improvement in quality of life by participation in lifelong learning—Economic stability | ① Not helpful at all, ② Not very helpful, ③ Medium, ④ Slightly helpful, ⑤ Very helpful, |

**Table 3.** Target (y).

| DQ9 | Employment type | ① Wage worker, ② Non-wage worker |
|-----|-----------------|----------------------------------|

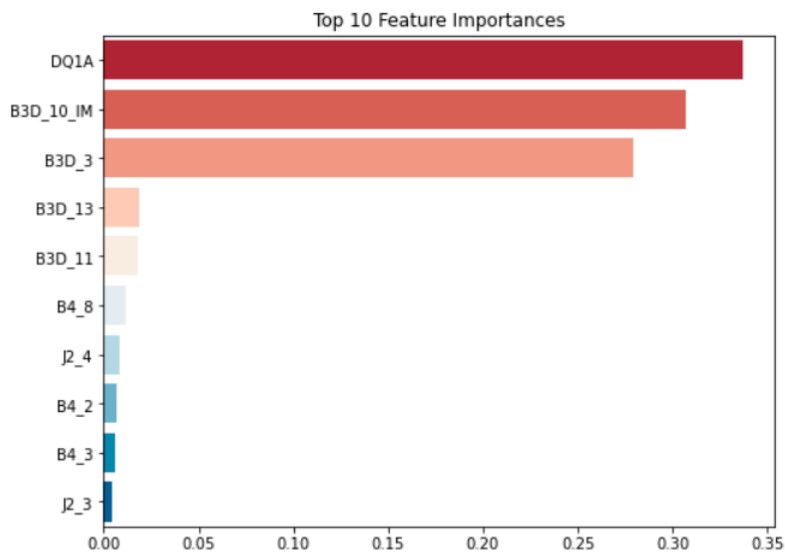

**Figure 4.** Ranking of important features through regression analysis.

## 3. Results

### 3.1. Gradient Boost Machine (GBM)

The importance of features as shown in Figure 5 was obtained by GBM for classification. Fundamentally, even if the default parameters provided by the python scikit-learn library are used without adjustment, good results can be obtained. The reason is that it was reconfirmed that the academic level and self-pay learning cost of the importance of each feature found in the basic regression analysis are the most important. Based on the target (① wage workers, ② non-wage workers), the confusion matrix as shown in Figure 6 was also confirmed. However, as mentioned in the introduction, this study intends to perform better hyperparameter tuning through grid search through user insight.

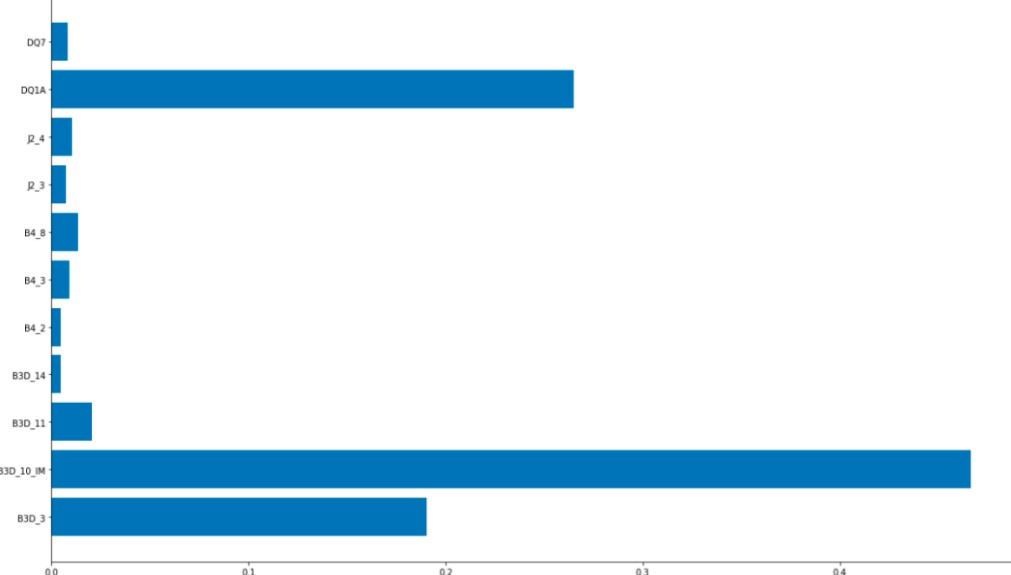

**Figure 5.** Feature importance analysis according to GBM.

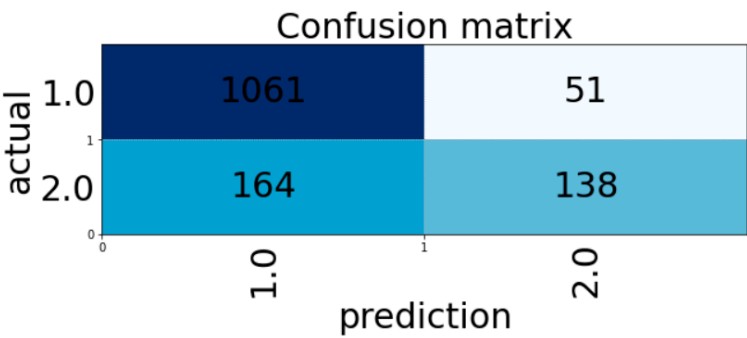

**Figure 6.** Confusion matrix of GBM.

Generally, the overall parameters of an ensemble model can be divided into three categories: (1) Tree-specific parameters which affect each individual tree of the model. (2) Boosting parameter which affect the boosting operation of the model. (3) Miscellaneous parameters which are other parameters for the entire function. This research focuses on the parameter tuning of (1) and (2). Figure 7 shows the general structure of a decision tree, so we will consider parameter tuning for a general decision tree first. This study uses Python's scikit-learn library, but even if R is used, the specific terminologies are slightly different, but the basic tuning idea is the same.

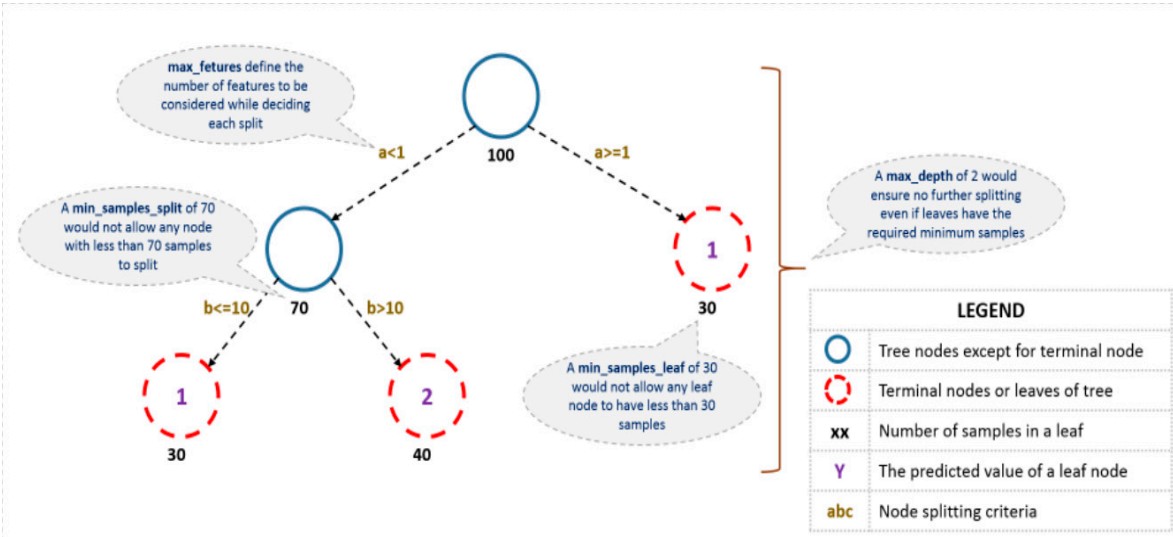

**Figure 7.** Traditional decision tree [32].

### 3.2. Grid Search

In this study, among the tree parameter options shown in Figure 8, 'min_samples_split': 150, 'min_samples_leaf': 40, 'max_depth': 7 and 'max_features': 'sqrt' were found as optimal parameter values. At this time, 'learning_rate' was 0.15 (usually 0.05–0.2) and n_estimators' was 50 (usually 40–70), so it was judged to be a reasonable optimal parameter. After fixing the tree-related parameters, the boosting parameter options as shown in Figure 9 were performed. After that, 'learning_rate': 0.15, 'n_estimators': 50, and 'subsample': 0.8 were found as optimal boosting parameter options. In the parameter combination of this study, 50 was obtained as 'n_estimators', an optimal estimate for 'learning_rate': 0.15. This is a fairly reasonable value and could be used as it is. However, since it may not be the same in all cases, it was run again, lowering the learning rate by 1/10 to 0.015 and raising the estimate tenfold from 50 to 500. Usually, in this way, the optimal parameters can be further improved, and then the parameters for each tree can be adjusted again. However, in this case, the order of the tuning variables must be carefully determined. That is, the variable

that has a greater influence on the outcome should be taken first. For example, max_depth and min_samples_split have a significant impact and should be carried out first. Finally, in this study, parameter tuning as shown in Figure 10 was performed. As a result of this study, it can be seen that better parameter tuning results can be obtained through more diverse combinations. This study confirmed that even if a basic library is used, parameter tuning based on user insight can show much better results.

'min_samples_split' : range(50,1001,50),

'min_samples_leaf' : range(30,71,10),

'max_depth' : [3,4,5,6,7,8,9],

'max_features' : ['sqrt']}]

**Figure 8.** Decision tree parameter options used for grid search.

'n_estimators':range(20,100,10),

'subsample':[0.8],

'learning_rate':[0.05, 0.1, 0.15],

**Figure 9.** Boosting parameter options used for grid search.

param_grid_search = [ {'n_estimators':[500],

'subsample':[0.8],

'learning_rate':[0.015],

'min_samples_split' : [150],

'min_samples_leaf' : [40],

'max_depth' : [7],

'max_features' : ['sqrt']}]

**Figure 10.** Final parameter values through tuning used for grid search.

After the grid search, the importance was re-confirmed in Figure 11. Although the highest level of education completed and self-pay education expenses are still important, it is confirmed that the difference in importance between the two is similar. Usually, it is better to have the high-priority features become similar. Figure 12 shows the confusion matrix with grid search. True positive rate was slightly lowered from 1061 to 1057, but true negative rate was raised from 138 to 143. In particular, false true rate improved from 164 to 159. Table 4 contains a comparison table of train scores and test scores.

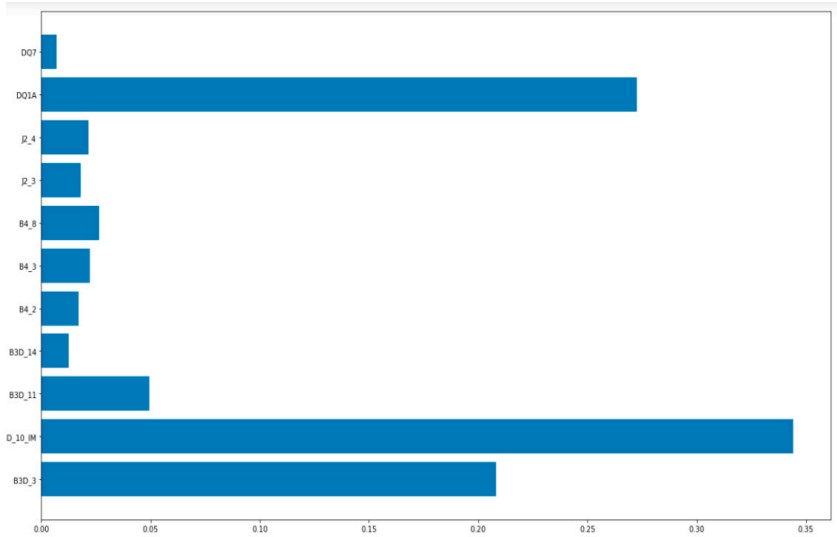

**Figure 11.** Improving importance through GBM grid search.

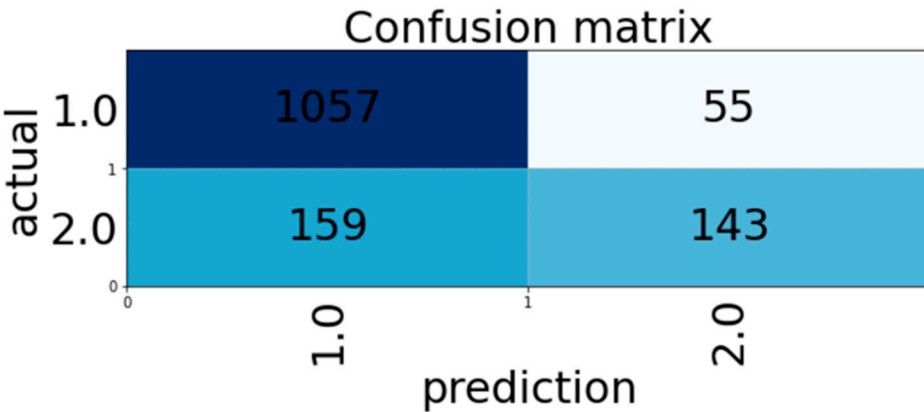

**Figure 12.** Confusion matrix through GBM grid search.

**Table 4.** Comparison table of train_score and test_score.

|  | Train_Score | Test_Score |
| --- | --- | --- |
| Gradient Boosting | 0.8598726114649682 | 0.847949080622348 |
| Grid Search | 0.8646496815286624 | 0.848656294200848 |

Table 5 compares overall accuracy, sensitivity, precision or positive predictive value, specificity (1-false positive rate), and f1_score. Overall, the numerical values have been improved, and it can be further improved through more diverse combinations of hyper-parameters. In particular, it can be seen that the false-positive rate (normal is judged as incorrect) has significantly improved. The desirable classification method should be highly sensitive and specific.

**Table 5.** Comparison report of gradient boosting and grid search.

|  | Gradient Boosting | Grid Search |
| --- | --- | --- |
| Overall accuracy | 0.84795 | 0.84866 |
| Sensitivity | 0.95414 | 0.95054 |
| Precision | 0.86612 | 0.86924 |
| **Specificity** | **0.45695364** | **0.47350993** |
| f1_score | 0.90800 | 0.90808 |

Specificity was significantly improved in the hyperparameter tuning of this study, showing how low the false negative rate (FNR; 1—specificity), which is the rate at which non-wage workers are judged to be wage workers. This suggests that tuning through grid search is a much better way to select non-wage workers who can benefit from regular gradient boosting. In addition, the specificity shows that the applicability of Bayes' Theorem can be considered in the decision-making of selection and classification. The most important criterion for judging the validity of a classification method is precision, which is the same as the posterior of Bayes theorem. Therefore, the results of this study suggest that a simpler Bayes theorem can be used for data handling similar to this study in place of the complex and time-consuming gradient boosting and grid search.

## 4. Discussion and Conclusions

In this study, the likelihood of Korean adults participating in lifelong education was predicted using GBM with grid search, a tree-based machine learning classification algorithm. The prediction was made using 11 independent variables such as program type, program satisfaction, and social participation. Among the variables used, the most

influential variable in predicting the possibility of lifelong education participation was self-pay education expenses and the next variable was the highest level of education completed. Both of these main predictor variables of self-pay education expenses and the highest level of education completed are limited to the micro level. In order to promote participation in lifelong learning, efforts should be made not only at the micro (individual) level but also at the meso (institutional) level and macro (socio-contextual) level from an ecological point of view. Particularly, the predictor of self-pay education expenses which are the first predictor of lifelong education intentions found through GBM with grid search has to do with Individual characteristics regarding the living (economic) situation. On the other hand, De Meester and other colleagues [35] presented financial costs such as program fees and course materials as the characteristics of the educational program at the meso level.

After the grid search, not only the importance of the two variables but also the overall figures improved. In particular, the false positive rate improved significantly. In this study, the performance of the GBM model was improved by using grid search. In general, hyper-parameters must be manually enhanced by the insight of the data processing specialists or an optimized algorithm must be developed for the manipulation. However, since the proposed method showed better results even with ready-made Grid-Search, it shows that users can acquire better results with an automated scientific method rather than a manual one. In addition, the results of the proposed method show that false-positive rate (1—specificity) in confusion matrix is the most improved. The important criteria for evaluating the validity of a classification are precision and false-positive rate (1-specificity), which are the same as the Bayesian theorem. Therefore, it is one of the contributions of the proposed method to show that the Bayesian theorem can be used less computationally instead of the complex and time-consuming GBM and grid search in research fields that require evaluation of results similar to this study. In future studies, it will be possible to improve the performance of the machine learning model by less computational methods. Furthermore, the results of this study can be applied to the prediction of participation in various learning forms (i.e., face-to-face learning, online learning, blended learning, and flipped learning) of K-12 and higher education in addition to lifelong education in the post-COVID-19 era. Also, it is expected to be used as a basis for big data research in the educational field.

**Author Contributions:** Conceptualization, T.P.; methodology, C.K.; validation, C.K. and T.P.; formal analysis, C.K. and T.P.; investigation, C.K.; resources, T.P; data curation, C.K. and T.P.; writing—original draft preparation, C.K.; writing—review and editing, C.K. and T.P.; supervision and project administration, T.P. All authors have read and agreed to the published version of the manuscript.

**Funding:** This work was supported by the Pukyong National University Research Fund in 2021 (CD20210841).

**Institutional Review Board Statement:** Not applicable.

**Informed Consent Statement:** Not applicable.

**Data Availability Statement:** The data used to support the findings of this study are included within the article.

**Acknowledgments:** Date for this study comes from the Seoul Education Research & Information Institute.

**Conflicts of Interest:** The authors declare no conflict of interest.

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
