# Peer review of "Predicting Determinants of Lifelong Learning Intention Using Gradient Boosting Machine (GBM) with Grid Search"

_sustainability, doi:10.3390/su14095256_

Round 1
Reviewer 1 Report
The overall goal of the manuscript was to explore the factors that influence actual learning intention leading to participation in adult education. For the development of the predictive model, the authors used tree-based machine learning with longitudinal data collected from the 2017 - 2020 longitudinal study of the Lifelong Learning Survey in Seoul, Republic of Korea.
The article from a methodological point of view meets the expectations of a scientific article. The authors establish clear categories of analysis and comprehensively explain the procedure used for both data collection and data analysis. The conceptual framework describing the theoretical background of factors influencing the intention to learn throughout life could perhaps be expanded to include citations from more recent authors. Reinforcement-based machine learning, on the other hand, is well explained and included as part of the theoretical framework.
The methodology is also very clear and well worked out from a practical point of view, however, the discussion and conclusions are insufficient. In some cases, what is mentioned in the summary is repeated verbatim without providing sufficient critical analysis and reasoned reflection of the results explained and obtained in the methodology section. Another recommended improvement is to increase the quality of the graphics. In particular, Figures 1, 2, 3 and 6 do not have sufficient image quality to be read clearly and included in the text as evidence of the analysis. It is recommended that the figures be processed and incorporated not as images but as graphics to avoid this type of problem. Once these changes have been incorporated, both in the theoretical framework and in the discussion, conclusions and figures, the article could be published in the Sustainability Journal.
Author Response
Dear Reviewer,
We would like to thank you for taking time to review our manuscript as well as to provide invaluable insights to enhance the depth of our research. Please refer to the attached file. The revised parts were marked in red. We have modified our manuscript according to your recommendations. They are shown in the table format as follows:
|
Reviewer 1 |
||
|
Reviewer's comments |
Revisions |
|
|
2. Theoretical Background
|
The conceptual framework describing the theoretical background of factors influencing the intention to learn throughout life could perhaps be expanded to include citations from more recent authors |
Park & Kim (2020), Thongmak (2021), and other recent references were added to revise the overall ‘2. Theoretical Background’.
Park, T.; Kim, C. Predicting the variables that determine university (re-) entrance as a career development using support vector machines with recursive feature elimination: The case of South Korea. Sus. 2020, 12, 7365.
Thongmak, M. Inquiring into lifelong learning intention: comparisons of gender, employment status, and media exposure. Int’l J. of Lifelong Edu. 2021, 40, 72-90.
Yağcı, M. Educational data mining: prediction of students' academic performance using machine learning algorithms. Smart Learning Environ. 2022, 9, 1-19.
|
|
5. Discussion and conclusions
|
The discussion and conclusions are insufficient. In some cases, what is mentioned in the summary is repeated verbatim without providing sufficient critical analysis and reasoned reflection of the results explained and obtained in the methodology section. |
Some repetitive or meaningless content was deleted from the ‘5. Discussion and conclusions’ part, and the implications of the analysis results were presented in a more sophisticated way. |
|
Miscellaneous |
Another recommended improvement is to increase the quality of the graphics. In particular, Figures 1, 2, 3 and 6 do not have sufficient image quality to be read clearly and included in the text as evidence of the analysis. It is recommended that the figures be processed and incorporated not as images but as graphics to avoid this type of problem. |
We increased resolution of Tables 1, 2, 3, and 6 and improved the readability of the text and image. |

Reviewer 2 Report
First of all, thank you for inviting me to review the paper entitled "Predicting Determinants of Lifelong Learning Intention Using GBM with Grid Search", very intriguing title at first glance, with longitudinal data should be very significant, the paper explored the factors that have the most decisive influence on actual learning intention that leads to participation in adult education - lifelong learning, which is a very interesting and significant topic.
However, as i read through the paper - i found that the paper is quite focused (too much) on the method and statistical analysis of the data, resulting to minimal discussion and implications of the finding, which is actually be the core contribution of the paper.
if method and statistical analysis is the focus - the paper might not be fitting for the "From Education and Humanities to Improve Knowledge, Society and the Digital Transformation" theme.
Additional recommendations are provided:
- GBM is Gradient Boosting Machine? - should be placed also in the abstract
- table the author noted the various micro, meso, and macro level, how are these reflected in the literature (from line 51 to 117), seems to be not that organized, would suggest to go over this section, since these three categories would be used in later analysis.
- section 2.2 - Boosting-based Machine Learning; this should be in the method section (or if within the literature - should be more focused on previous studies using GBM
- line 246 - data sets used - which information are classified and used for the micro, meso, and macro level? Was regression used?
- then the authors just went on providing the results of GBM, which is too overwhelming - should be simplify
- Discussion and conclusion is too weak - should go back to the theoretical concepts and relate the findings
was hoping to find some implications on how self-pay education expenses and highest educational attainment were supposed to be the predictors for life long learning
In sum, the paper do have some potential, but should shift its focused on the relevant sustainable issues related to the topic / theme and not too much on the method and statistics
Author Response
Dear Reviewer,
We would like to thank you for taking time to review our manuscript as well as to provide invaluable insights to enhance the depth of our research. Please refer to the attached file. The revised parts were marked in red. We have modified our manuscript according to your recommendations. They are shown in the table format as follows:
|
Reviewer 2 |
||
|
Reviewer's comments |
Revisions |
|
|
Overall |
However, as i read through the paper - i found that the paper is quite focused (too much) on the method and statistical analysis of the data, resulting to minimal discussion and implications of the finding, which is actually be the core contribution of the paper. if method and statistical analysis is the focus - the paper might not be fitting for the "From Education and Humanities to Improve Knowledge, Society and the Digital Transformation" theme. In sum, the paper do have some potential, but should shift its focused on the relevant sustainable issues related to the topic / theme and not too much on the method and statistics
|
Throughout the manuscript, the description of statistical methods has been simplified and revised so that the research focus can be on lifelong learning (education). |
|
Title, Abstract |
GBM is Gradient Boosting Machine? - should be placed also in the abstract |
In the title and abstract, GBM is presented as a Gradient Boosting Machine (GBM). |
|
2. Theoretical Background 2.1. Factors affecting the lifelong learning intention |
table the author noted the various micro, meso, and macro level, how are these reflected in the literature (from line 51 to 117), seems to be not that organized, would suggest to go over this section, since these three categories would be used in later analysis.
|
The literature review (Line 119 to 144) was modified to clearly explain micro-, meso-, and macro-levels of the environment of surrounding learners’ lifelong learning intentions. Also, these ecological levels were linked to the finding and discussion and conclusion. |
|
2.2. Boosting-based Machine Learning |
Boosting-based Machine Learning; this should be in the method section (or if within the literature - should be more focused on previous studies using GBM |
To accommodate your kind recommendation, we moved ‘2.2. Boosting-based Machine Learning’ to ‘3.1. Boosting-based Machine Learning’ in method section. |
|
3. Methods |
data sets used - which information are classified and used for the micro, meso, and macro level? Was regression used? |
The questions presented in Table 2 were restructured by dividing them into micro-, meso-, and macro-level of factors influencing lifelong learning from ecological perspectives. |
|
4. Results |
then the authors just went on providing the results of GBM, which is too overwhelming - should be simplify
|
In the result part of GBM, content not directly related to this study (ex. the description of three categories of parameters of an ensemble model) was deleted. |
|
5. Discussion and conclusions |
Discussion and conclusion is too weak - should go back to the theoretical concepts and relate the findings was hoping to find some implications on how self-pay education expenses and highest educational attainment were supposed to be the predictors for life long learning |
In the ‘4. Discussion and conclusions’ part, we added the content of how the main predictors (self-pay education expenses and the highest level of education completed) affecting the intention to participate in lifelong education are interpreted and what implications they have according to the three levels suggested from the ecological point of view (Line 394-402), and the valuable meaning of the GBM and grid search methodologies (Line 405-417). |

Round 2
Reviewer 2 Report
Thank you for inviting me to review the revised version of this paper, evident of revisions are reflected within the paper.
However, some minor issues are still noted.
- please recheck the numbering of the figures. in line 355 - "After the grid search, the importance was re-confirmed in Figure 10", which is actually figure 11.
- similar errors are also found in the next figure. Author please recheck all the figures.
- line 415 - time-consuming gradient boosting (GBM) - this might be mistaken as an acronmym - do you mean "....time-consuming gradient boosting or GBM" or "....time-consuming gradient boosting/GBM" , please recheck.
Author Response
Dear Reviewer,
We appreciate your feedback as it gave us the opportunity to improve the quality of the manuscript. Please refer to the attached file. The revised parts were marked in red.
